# Cerebral Blood Flow in Healthy Subjects with Different Hypnotizability Scores

**DOI:** 10.3390/brainsci12050558

**Published:** 2022-04-26

**Authors:** Anas Rashid, Enrica Laura Santarcangelo, Silvestro Roatta

**Affiliations:** 1Lab of Integrative Physiology, Department of Neuroscience “Rita Levi Montalcini”, University of Torino, 10125 Torino, Italy; anas.rashid@unito.it (A.R.); silvestro.roatta@unito.it (S.R.); 2Lab of Cognitive and Behavioral Neuroscience, Department of Translational Research and New Technologies in Medicine and Surgery, University of Pisa, 56127 Pisa, Italy

**Keywords:** TCD, NIRS, hyperventilation, rebreathing, cognitive tasks, hypnotic susceptibility

## Abstract

Hypnotizability is a cognitive trait associated with differences in the brachial artery flow-mediated dilatation of individuals with high hypnotizability (highs) and low hypnotizability scores (lows). The study investigated possible hypnotizability-related cerebrovascular differences. Among 24 healthy volunteers, the Stanford Hypnotic Susceptibility Scale Form A identified 13 medium-to-lows (med-lows), 11 medium-to-highs (med-highs), and 1 medium hypnotizable. Hypnotizability did not influence the significant changes produced by the trail making task (TMT), mental arithmetic task (MAT), hyperventilation (HVT), and rebreathing (RBT) on heart rate (HR), arterial blood pressure (ABP), and partial pressure of end-tidal CO_2_ (P_ET_CO_2_), but moderated the correlations between the changes occurring during tasks with respect to basal conditions (Δ) in ABP and P_ET_CO_2_ with middle cerebral artery flow velocity (MCAv). In HVT, med-lows exhibited a significant correlation between ΔMCAv and ΔP_ET_CO_2_, and med-highs showed a significant correlation between ΔABP and ΔMCAv. Cerebrovascular reactivity (CVR) and conductance (ΔCVCi) were significantly correlated with ΔMCAv only in med-lows during HVT and RBT. For the first time, cerebrovascular reactivity related to hypnotizability was investigated, evidencing different correlations among hemodynamic variables in med-highs and med-lows.

## 1. Introduction

Hypnotizability—the proneness to modify perception, memory, and behavior according to specific suggestions and to enter a hypnotic state [1]—is a psychophysiological trait [2] measured by scales whose scores are stable during adulthood [3]. Individuals with high- (highs) and low-hypnotizability scores (lows) display different morpho-functional brain characteristics mainly concerning the salience and executive networks [4] and the cerebellum [5]. They also exhibit differences in the cognitive, sensorimotor, and cardiovascular domains observable in the absence of hypnotic induction and suggestions, thus being relevant to everyday life [2]. For instance, with respect to lows and medium-hypnotizable individuals (mediums), highs show a greater ability in respect of attention and dissociation [6,7,8], stronger functional equivalence between imagery and perception [9,10], low interoceptive accuracy [11], greater interoceptive sensitivity [12], and less close postural and visuomotor control [2].

In the vascular system, the brachial artery post occlusion flow-mediated dilatation (FMD) is similar in highs and lows in basal conditions, whereas FMD is less impaired in highs than in lows during nociceptive stimulation [13] and is not impaired at all in highs during mental stress, at variance with lows and the general population [14,15]. This can be accounted for by different control of the release of nitric oxide (NO) by endothelial cells [16] in highs and lows/general population during cognitive and physical stimulation.

Hypnotizability-related differences in the cerebral artery blood flow are possibly due to different NO availability, which has not been studied. They could be relevant to the observed negative correlation between hypnotizability and brain total grey matter volume [4,5,8]. NO is required, in fact, for the development and maturation of the nervous tissue, but its excessive amount can be toxic, mainly for regions particularly rich in NO such as the cerebellum and hippocampus [17].

### 1.1. Changes in the Cerebral Artery Diameter Induced by Hypercapnia/Hypocapnia and Arterial Pressure

In the brain tissue, perfusion, or cerebral blood flow (CBF), is a measure of the rate of delivery of arterial blood to a capillary bed [18]. Hypocapnia is associated with reduced CBF [19], NO availability [20], and artery diameter [21], not depending on extracellular pH [22]. It seems unlikely that the formation of NO is the only mechanism involved in hypercapnia, because increases in CBF during very high levels of hypercapnia (>100 mmHg) are not altered by the inhibition of NO synthase (NOS). In contrast, vasodilation during moderate hypercapnia appears to be dependent on the production of NO due to increased activity of NOS [23]. Another possibility is that NO is not the direct mediator of vessel relaxation, but that normal basal levels of NO and/or cyclic guanosine monophosphate (cGMP) are required for the response to hypercapnia to occur [24].

The hypothesis that artery dilation may be mediated by a decreased pH in the extracellular fluid, as CO_2_ freely diffuses across the cerebrovascular endothelium, is contradicted by experiments reporting the effects of blockade of the cerebrovascular action of CO_2_ by brainstem lesions [25,26] or decerebration [27,28,29], cholinergic blockers [30,31,32,33], inhibitors of prostaglandin [34,35], and NO synthesis [36].

Cerebrovascular changes induced by increased or decreased CO_2_ cannot be reliably studied without concomitant assessment of arterial blood pressure [37]. It has also been shown, however, that NO release is not obligatory for cerebrovascular reactivity to CO_2_, whereas transient increases in CBF induced by increases in blood pressure mostly rely on NO release [38]. Cerebral autoregulation, in fact, stabilizes blood flow by buffering the steady state increase in cerebral perfusion pressure, but cannot compensate for brief elevations in blood pressure and consequent increases in CBF [39,40].

### 1.2. Changes in the Cerebral Artery Diameter Induced by Brain Activity

NO mediates the brain vessel dilation, which follows increased neuronal activity leading to the increased need for O_2_ supply [41,42]. In this perspective, we may expect that hypnotizability-related differences in the cerebral artery diameters occur during cognitive tasks. The highs’ greater proneness to focused attention and absorption [6,43], in fact, might enable them to activate task-related areas more than lows’ and, thus, exhibit a larger increase in CBF. However, the highs’ greater attentional abilities [43,44,45] may require less neural engagement, as observed, for instance, in motor imagery tasks performed by individuals with greater motor experience of those tasks [46]. In addition, the highs’ and lows’ modes of information processing—that is, their neural activities—is different. During sensory and cognitive tasks, the former display slight and diffuse changes in the general asset of brain activation, whereas the latter exhibit more localized task-related changes [9,47,48]. This may lead to negligible differences in the highs’ brain metabolic changes during tasks with respect to basal conditions. Thus, we cannot reliably predict whether and how hypnotizability modulates cerebral hyperemia induced by cognitive tasks.

### 1.3. Aims of the Study

The study aimed to assess (a) whether the flow velocity in the middle cerebral artery (MCAv) is influenced by hypnotizability during cognitive tasks, hyperventilation, and rebreathing; and (b) how MCAv is associated with the partial pressure of end-tidal CO_2_ and arterial blood pressure (ABP) in healthy subjects with different hypnotizability.

## 2. Materials and Methods

### 2.1. Subjects

A total of 24 healthy subjects (age: 26.1 ± 4.5 years; 12 females and 12 males), 20 right-handed and 4 left-handed according to the Edinburgh Handedness Inventory [49], were enrolled in this study. They were all university students with no medical, neurological, and psychiatric disease; hypertension (resting blood pressure <120/80 mmHg); sleep and attention disturbance; substance abuse throughout their life; or drug intake in the last 3 months, as self-reported while signing the informed consent.

### 2.2. Experimental Procedure

The hypnotic assessment was performed by the validated Italian version of the Stanford Hypnotic Susceptibility Scale (SHSS), Form A [50], which is a behavioral scale classifying highs (score: 8–12 items passed out of 12), mediums (score: 5–7 out of 12), and lows (score: 0–4 out of 12). It includes motor and cognitive items indicating motor inhibition, hallucination, and dissociation abilities and its administration requires approximately 20 min. It was administered to all participants by the same expert experimenter. Owing to the small number of participants, they were divided into two groups: medium-low (SHSS score: 0–5) and medium-high (SHSS score: 7–12) hypnotizable.

Experiments were conducted in a quiet, sound and light attenuated, temperature-controlled (21–23 °C) room between 9 and 11 AM, and 4 and 6 PM, at least 3 h after the latest food and caffeine/alcohol intake. Before starting the recordings, the partial pressure of end-tidal CO_2_ (P_ET_CO_2_) was measured and the signal visual feedback was provided to the participants, who were invited to relax sitting in an armchair. After all signals reached a stable condition, i.e., at least for 5 min, they were recorded for a 10 min baseline period and during a sequence of four tasks separated from each other by a minimum of a 5 min rest (with additional time if required by the participant) and randomly presented within med-lows and med-highs.

Since hypnotizability is a dispositional trait that is substantially stable through life [3], the recordings performed 2 months earlier than hypnotic assessment could be reliably studied as a function of hypnotizability.

### 2.3. Tests

Before starting the test sequence, participants were briefly familiarized with the tasks and respiratory maneuvers, particularly with the visual feedback during hyperventilation (described below). Each test consisted of a basal and a task condition. The test sequence was randomized for the different subjects.

#### 2.3.1. Trail Making Task (TMT)

The participant was instructed to connect numbers (1 to 40) and 21 Italian letters (A to Z) by lines in ascending sequence, which was performed on paper over 3 min with alternating numbers and letters (1–A–2–B–3–C and so on).

#### 2.3.2. Mental Arithmetic Task (MAT)

The participant was asked to progressively subtract odd numbers (1, 3, 5, and so on) from 1000 for 3 min, writing each result on a paper. One operator standing behind the subject monitored the outcome, promptly asking the subject to repeat the calculation in case of error.

After TMT and MAT, the participant was invited to rate the experienced cognitive fatigue using a numerical rating scale from 0 (min) to 10 (max).

#### 2.3.3. Hyperventilation (HVT)

The participant was asked to hyperventilate to achieve and maintain for 90 s 50% of the normal P_ET_CO_2_. To this end, they were provided with visual feedback from the display of the capnograph, which was continuously monitoring P_ET_CO_2_ from the expiratory flow collected by a nasal cannula and a horizontal cursor placed at 50% P_ET_CO_2_ indicating the target P_ET_CO_2_.

#### 2.3.4. Rebreathing (RBT)

The nose was closed with the help of a nose clip. The participant was asked to take a large breath of room air and then exhale into a previously empty plastic bag, closing the bag so that it stayed full. Once the bag was full of expired air, the participant resumed normal breathing in and out of the closed bag, hence maintaining a spontaneous breathing frequency, and when the participant began to rebreathe, we started a stopwatch. The participant continued to rebreathe for 90 s. The mouthpiece was also connected with a sampling line to the capnograph to allow continuous recording of P_ET_CO_2_.

### 2.4. Measurements

The heart rate (HR, bpm) and arterial blood pressure (ABP, mmHg) were measured by continuous finger-pulse photoplethysmography (CNAP Monitor 500, CNSystems Medizintechnik GmbH, Graz, Austria). Calibration of ABP was periodically performed using a regular pneumatic cuff at the right arm.

A capnograph (Capnostream™ 20p Bedside Patient Monitor with Microstream™ Technology, Oridion Medical, Jerusalem, Israel) was used to monitor the partial pressure of end-tidal carbon dioxide (P_ET_CO_2_) in the respiratory gases.

Transcranial Doppler (TCD) ultrasound with a 2 MHz monitoring probe (Dolphin IQ and 4D, Viasonix, Netanya, Israel) was used to measure unilateral cerebral flow velocity from the middle cerebral artery (MCAv, cm/s) during all tasks. The probe was held in place by a 3D-printed custom-made helmet.

Cerebrovascular reactivity (CVR, cm/s/mmHg) was computed as a ratio between task-related changes in MCAv and P_ET_CO_2_ (ΔMCAv/ΔP_ET_CO_2_), and cerebrovascular conductance index (CVCi, cm/s/mmHg) was computed as a ratio between MCAv and ABP. ΔCVCi was computed as ([MCAv/ABP]_t_ − [MCAv/ABP]_b_), where t and b represent task and basal values, respectively.

All signals were continuously digitally sampled (CED Micro 1401, Cambridge Electronic Design Ltd., Cambridge, UK) at 100 Hz and stored on a computer. Spike2 software (Version 9.14, Cambridge Electronic Design Ltd., Cambridge, UK) was used for both data acquisition and analysis.

### 2.5. Statistical Analysis

Statistical analysis was performed using MATLAB^®^ Version R2022a (The MathWorks, Natick, MA, USA). After normality assessment (Kolmogorov–Smirnov test), the entire sample HR, ABP, P_ET_CO_2_, and MCAv were submitted to repeated-measures ANOVA according to a 4 test (basal(b)-TMT, b-MAT, b-HVT, b-RBT) × 2 condition (b, task) experimental design. Post hoc analyses were performed through paired t-tests. Since sample size did not allow a reliable comparison between groups, ANOVA was repeated using SHSS scores as a covariate.

For the tasks showing changes in MCAv with respect to basal conditions, Spearman coefficients able to reveal nonlinear correlations were computed between the averaged basal values of MCAv, ABP, and P_ET_CO_2_. Assuming that ABP and P_ET_CO_2_ were the systemic variables influencing MCAv, the changes in MCAv (ΔMCAv), ABP (ΔABP), and P_ET_CO_2_ (ΔP_ET_CO_2_) occurring during tasks with respect to basal conditions were computed for the entire sample. The correlations of ΔMCAv with ΔABP and ΔP_ET_CO_2_, and with the derived variables, i.e., CVR and ΔCVCi, were computed. Partial correlations controlling for hypnotizability were also performed, and successive within-group correlations were studied when the entire sample partial correlations revealed a moderation by hypnotizability. For all analyses, significance was set at *p* = 0.05.

## 3. Results

Hypnotic assessment identified 11 lows (SHSS mean score ± standard deviation (SD); 2 ± 1.58), 5 mediums (6 ± 1), and 8 highs (8.5 ± 0.71). Excluding the subject with SHSS score = 6, the sample included 13 medium-lows (med-lows; SHSS range: 0–5; mean score ± SD: 1.38 ± 1.98) and 10 medium-highs (med-highs; SHSS range: 7–12; mean score ± SD: 8.1 ± 0.73).

On a numerical rating scale ranging from 0 (minimum) to 10 (maximum), the experienced cognitive fatigue during TMT and MAT tasks considered together (Mean ± SD) was 6.5 ± 0.63 (med-lows: 6.27 ± 0.63; med-highs: 6.75 ± 0.54).

Systemic and Doppler variables exhibited different changes during the four tasks with respect to basal conditions, which were not influenced by hypnotizability (repeated-measures ANOVA). Correlational analysis, in contrast, revealed hypnotizability-related differences in the correlation of MCAv with ABP (which was significant only in med-highs) and with P_ET_CO_2_ (which was significant only in med-lows); these were the main findings of the study, together with the observation that ABP and P_ET_CO_2_ jointly control CVR and ∆CVCi only in med-lows. Table 1 reports mean values and standard deviations of HR, ABP, P_ET_CO_2_, and MCAv.

### 3.1. Differences between Med-Highs and Med-Lows

#### 3.1.1. Systemic Measures

Systemic variables did not exhibit significant difference between the two groups. In detail: HR exhibited a significant test x condition interaction (*F*(3,69) = 7.511, *p* = 0.002, *η^2^* = 0.246, *α* = 0.982) whose decomposition revealed significant increases during tasks for all tests (TMT, *t* = 4.561; MAT, *t* = 4.895; HVT, *t* = 5.048 and RBT, *t* = 8.306; *p* = 0.0001). It survived after controlling for SHSS scores and was sustained by larger values during HVT than during MAT (*t* = 2.138, *p* = 0.043) and TMT (*t* = 3.242, *p* = 0.004).

For ABP, ANOVA revealed a significant test x condition interaction (*F*(3,69) = 218.26, *p* = 0.0001, *η^2^* = 0.905, α = 1.00), surviving after controlling for SHSS scores. Its decomposition showed significant differences between basal and task conditions for TMT (*t*(1,23) = 3.27, *p* = 0.003), MAT (*t* = 6.47, *p* = 0.0001), and RBT (*t* = 4.25, *p* = 0.0001), but not for HVT.

P_ET_CO_2_ exhibited a significant test x condition interaction (*F*(3,69) = 566.28, *p* = 0.0001, *η^2^* = 0.961, α = 1.00), which remained significant after controlling for SHSS scores. Decomposition revealed significant basal vs. task differences for HVT (*t* = 24.18, *p* = 0.0001) and RBT (*t* = 18.73, *p* = 0.0001) only.

Average response curves for hyperventilation (HVT) and rebreathing (RBT) with standard deviation for P_ET_CO_2_, ABP, and MCAv are shown in Figure 1.

#### 3.1.2. Doppler Measures

Doppler variables did not exhibit significant differences between the two groups. In detail: MCAv exhibited a significant test x condition interaction (*F*(3,69) = 217.48, *p* = 0.0001, *η^2^* = 0.904, α = 1.00), which remained significant considering SHSS as a covariate. As shown in Figure 2, its decomposition revealed a significant difference between basal and task conditions for HVT *t*(1,23) = 24.91, *p* = 0.0001) and RBT (*t* = 12.58, *p* = 0.0001) only.

For the derived variable ΔCVCi, the significant test × condition interaction (*F*(3,69) = 73,30, *p* = 0.0001, *η^2^* = 0.761, α = 1.00), which survived after controlling for hypnotizability, was sustained by a significant difference between basal and task conditions in all tests (TMT, *t* = 2.93, *p* = 0.008; MAT, *t* = 3.51, *p* = 0.002; HVT, *t* = 16.48, *p* = 0.0001; and RBT, *t* = 6.98, *p* = 0.0001).

### 3.2. Associations between Systemic and Doppler Variables

Since MCAv did not exhibit significant differences between basal and experimental conditions in TMT and MAT, further analyses were conducted only on HVT and RBT.

#### 3.2.1. Hyperventilation (HVT)

Spearman coefficients computed for ΔMCAv, ΔABP, and ΔP_ET_CO_2_ for the entire sample indicated a significant correlation between ΔMCAv and ΔP_ET_CO_2_ (*ρ* = 0.484, *p* = 0.017), which became non-significant after removing the effects of hypnotizability by partial correlation. Successive within-group analyses revealed a significant correlation between ΔMCAv and ΔP_ET_CO_2_ (*ρ* = 0.606, *p* = 0.028) and disclosed a significant correlation of ΔMCAv with ΔABP (*ρ* = 0.842, *p* = 0.002) in med-highs (Figure 3).

In the entire sample, the significant correlations of ΔMCAv and CVR (*ρ* = −0.533, *p* = 0.007) and ΔCVCi (*ρ* = 0.598, *p* = 0.002) became non-significant after removing the effects of hypnotizability. Indeed, only med-lows exhibited significant correlations of ΔMCAv with CVR (*ρ* = −0.602, *p* = 0.020) and ΔCVCi (*ρ* = 0.861, *p* = 0.0001).

CVR and ΔCVCi mean values and standard deviations are reported in Table 2.

#### 3.2.2. Rebreathing (RBT)

No significant correlation between ΔMCAv with ΔP_ET_CO_2_ and ΔABP was observed in the entire sample and was disclosed after removing the effects of hypnotizability (Figure 4).

ΔMCAv was significantly correlated with CVR (*ρ* = 0.818, *p* = 0.0001), but not with ΔCVCi, and the correlation survived after removing the effects of hypnotizability (*ρ* = 0.629, *p* = 0.001).

## 4. Discussion

Our study was motivated by the observation of different brachial artery post-occlusion FMD in lows and highs. In the latter, FMD was scarcely and not influenced by nociceptive stimulation and mental stress, respectively, in contrast to lows who behaved like the general population [13,14], displaying decreased FMD. We wondered whether hypnotizability-related differences may also occur in the cerebrovascular responses to physical and mental stimulation.

### 4.1. Comparisons between Groups

The absence of significant changes in blood flow velocity during cognitive tasks in both groups contrasts with other authors’ findings reporting increased blood flow with increasing cognitive load in the general population [51]. However, Csipo et al. employed a visual text inspection task with high time pressure (a new task every 2 s); the methodological differences could account for different results. In addition, the discrepancy may be due to the characteristics of the studied sample, which does not reflect the distribution of hypnotizability in the general population [52]. The absence of increases in the middle cerebral artery blood flow, despite significant increases in systemic blood pressure, can be accounted for by the pre-eminent local control of cerebral blood flow. The similar results of med-lows and med-highs could be due to their different cognitive abilities, as med-lows may have paid poor attention to the tasks, and med-highs may have experienced low cognitive effort [9,43,45]. Thus, scarce neural engagement and, consequently, scarce increase in metabolism may have taken place in both groups.

The present findings confirm that, in the entire sample, hyperventilation and rebreathing induced a decrease and increase in middle cerebral artery flow velocity [53]. These responses occurred in the presence of significant changes in ABP only during rebreathing and in P_ET_CO_2_ during both rebreathing and hyperventilation and were not moderated by hypnotizability.

### 4.2. Association between Systemic and Doppler Variables

Even in the absence of significant differences between med-lows and med-highs in the mean values of ABP, P_ET_CO_2_, and MCAv, however, the task-related changes in these variables were differentially associated between each other in the two groups. This suggests different control mechanisms of blood flow velocity in med-lows and med-highs. During HVT, in fact, within-group correlational analyses revealed different relevance of the chemical (P_ET_CO_2_) and mechanical stimulation (ABP) to the med-lows’ and the med-highs’ cerebral blood flow velocity.

The absence of significant ABP changes during HVT does not allow us to exclude that the observed non-significant ABP decreases may have influenced the med-highs’ changes in cerebral blood flow ΔMCAv. An opposite behavior occurred in med-lows, who exhibited a significant association between ΔMCAv and ΔP_ET_CO_2_, indicating a preferential control of MCAv by chemical stimulation.

Theoretically, the decrease in MCAv observed in med-highs during HVT, despite scarce changes in ABP, could be sustained by more sensitivity of their vessel muscle cells to blood pressure. The effect of NO on precapillary sphincters and pericytes [54] may have contributed to the observed changes in MCAv.

The lack of significant correlation between ΔMCAv and ΔP_ET_CO_2_ occurring in both groups during RBT may be due to CO_2_ ceiling effects.

During both hyperventilation and rebreathing, an association of cerebrovascular reactivity (CVR, related to P_ET_CO_2_) and cerebrovascular conductance index (CVCi, related to ABP) was found only in med-lows. We hypothesize that a different interaction between arterial blood pressure and CO_2_ [37] occurred in med-highs and med-lows, respectively, owing to the presence of different local metabolic conditions depending on different modes of information processing [9].

## 5. Limitations and Conclusions

This study has a few limitations. One is the small sample size, which did not allow a direct comparison between highs, lows (each representing 15% of the general population), and mediums, which represent its largest part (70%) [52]. In a few tasks, in fact, highs or lows can behave like mediums [12]. Another limitation is the absence of very highly hypnotizable persons in the studied sample (the maximum SHSS score was 9 out of 12). Thus, the present study should be repeated in a larger sample including lows, mediums, and highs. Moreover, hemogas analyses may reflect blood CO_2_ better than end-tidal measures. Finally, given the complexity of the cerebrovascular control, recent approaches based on the complexity of the arterial pressure and cerebral blood flow time series may allow better detection of relationships between these two variables [55].

The present findings do not allow us to exclude the possibility that med-highs may exhibit a different association between ABP, P_ET_CO_2_, and CBF with respect to med-lows also when ABP increases instead of decreasing as occurs during HVT. In this respect, it may be interesting that more frequent increases in systemic blood pressure are likely to occur in highs than in lows owing to their greater emotional intensity, empathy [56,57], and interoceptive sensitivity [12,56]. In addition, changes in synaptic glutamate due to tasks could induce increases in astrocyte calcium-mediated dilatory action independently from systemic blood pressure [58]. Since highs respond to cognitive tasks with larger glutamatergic cortical activity [59], they might be more prone than lows to undergo astrocyte-dependent NO release in everyday life. Thus, the hypothesis that the earlier-observed highs’ reduced cerebral grey matter volume [4,5] may be due to the toxic effects of too much NO availability during environmental stimulation cannot be excluded.

In conclusion, this is the first assessment of hypnotizability-related differences in the mechanisms controlling cerebral blood flow, cerebrovascular reactivity, and conductance in response to hyperventilation and rebreathing, given the med-highs preferential response to arterial blood pressure, the med-lows apparently better response to CO_2_, and the different interaction between cerebrovascular conductance and reactivity in the two groups. The findings support the view that hypnotizability is associated with physiological correlates influencing everyday life independently from suggestions and induction of the hypnotic state [2,60]. Finally, it is noticeable that at variance with highs and lows, med-highs and med-lows represent half of the general population each. Thus, in a general perspective, the cerebrovascular reactivity of half of the population is expected to be more sensitive to changes in systemic blood pressure and the other half to the local increases in CO_2_ following the changes in blood pressure. Whether the different vascular reactivity to ABP changes of med-highs and med-lows is related to the different involvement of NO in the regulation of the vascular response remains to be ascertained. It may be speculated that this difference could influence the capacity to adapt to cerebrovascular insults.

## Figures and Tables

**Figure 1 brainsci-12-00558-f001:**
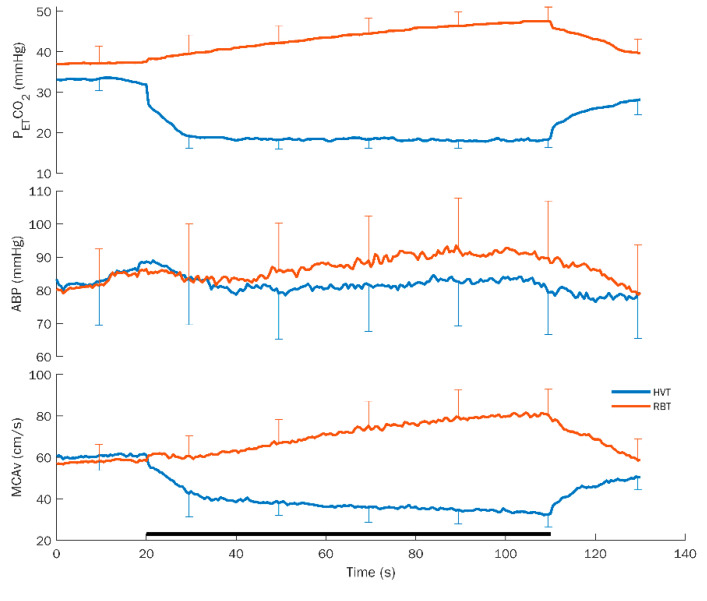
Average response curves for hyperventilation (HVT) and rebreathing (RBT) with standard deviation for the different variables. P_ET_CO_2_: partial pressure of end-tidal CO_2_; ABP: arterial blood pressure; and MCAv: middle cerebral artery flow velocity. The black bar at the bottom indicates the duration of the task. Note the opposite effects on each variable exhibited by the HVT and RBT.

**Figure 2 brainsci-12-00558-f002:**
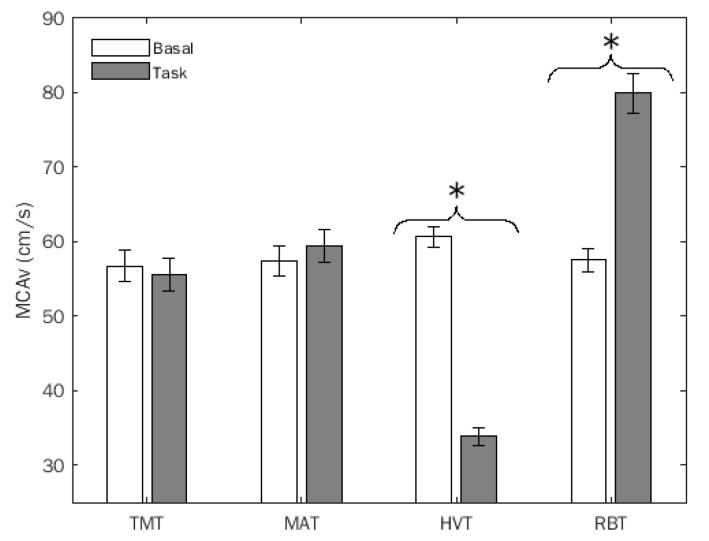
Middle cerebral artery flow velocity (MCAv) (mean, SEM) of the entire sample. TMT: trail making task; MAT: mental arithmetic task; HVT: hyperventilation; RBT: rebreathing. *, statistically significant differences.

**Figure 3 brainsci-12-00558-f003:**
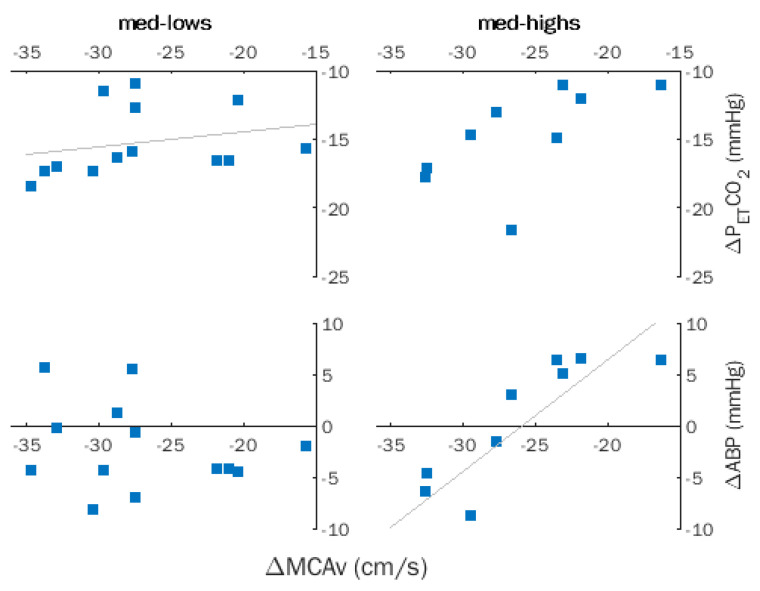
Hyperventilation (HVT). Within-group correlations between changes in middle cerebral artery flow velocity (MCAv), partial pressure of end-tidal CO_2_ (P_ET_CO_2_), and arterial blood pressure (ABP) in med-lows and med-highs. Trendlines indicate significant correlations.

**Figure 4 brainsci-12-00558-f004:**
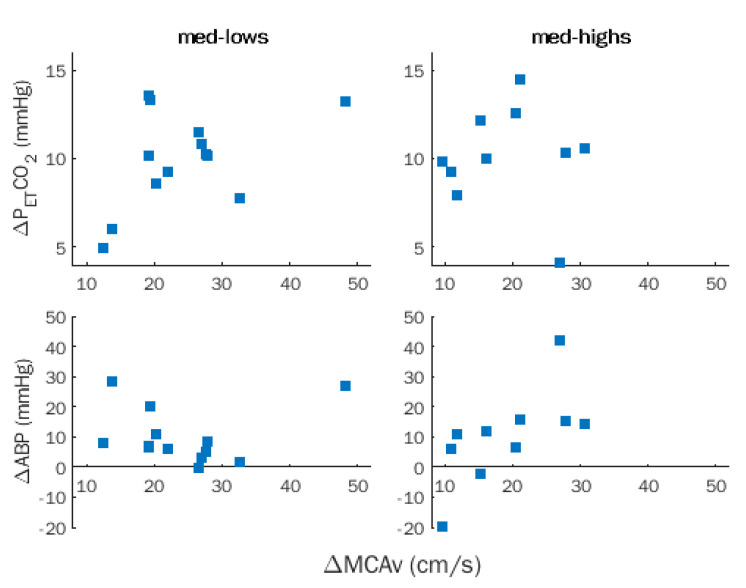
Rebreathing (RBT). Within-group correlations between changes in middle cerebral artery flow velocity (MCAv), partial pressure of end-tidal CO_2_ (P_ET_CO_2_), and arterial blood pressure (ABP) in med-lows and med-highs.

**Table 1 brainsci-12-00558-t001:** Variable mean values and standard deviations.

Condition	Variable	Med-Lows		Med-Highs	
*TMT*		Mean	SD	Mean	SD
Basal	HR (bpm)	72.02	12.08	78.50	14.66
	ABP (mmHg)	82.29	17.23	80.71	14.66
	P_ET_CO_2_ (mmHg)	37.31	1.91	34.63	3.45
	MCAv (cm/s)	59.52	9.61	52.13	10.43
Task	HR * (bpm)	79.09	13.66	81.62	14.46
	ABP * (mmHg)	87.95	14.72	86.56	13.98
	P_ET_CO_2_ (mmHg)	37.13	1.40	34.46	3.43
	MCAv (cm/s)	59.02	10.10	50.18	9.87
* **MAT** *					
Basal	HR (bpm)	75.98	11.61	76.50	14.10
	ABP (mmHg)	78.19	14.59	83.71	15.15
	P_ET_CO_2_ (mmHg)	36.51	1.92	34.53	3.29
	MCAv (cm/s)	60.25	8.70	52.56	10.61
Task	HR * (bpm)	84.20	12.41	84.51	15.00
	ABP * (mmHg)	88.64	12.91	96.89	9.80
	P_ET_CO_2_ (mmHg)	36.96	2.61	34.91	3.68
	MCAv (cm/s)	61.60	10.27	55.66	11.87
* **HVT** *					
Basal	HR (bpm)	72.93	10.09	77.88	11.36
	ABP (mmHg)	85.65	12.47	83.03	14.77
	P_ET_CO_2_ (mmHg)	33.61	2.49	32.21	3.73
	MCAv (cm/s)	61.25	7.24	59.68	6.67
Task	HR * (bpm)	99.34	19.71	84.12	13.01
	ABP (mmHg)	83.58	12.93	82.53	10.65
	P_ET_CO_2_ * (mmHg)	18.34	1.46	17.92	1.59
	MCAv * (cm/s)	34.19	7.07	33.35	5.17
* **RBT** *					
Basal	HR (bpm)	74.26	9.36	76.77	11.24
	ABP (mmHg)	78.35	12.82	85.23	11.25
	P_ET_CO_2_ (mmHg)	38.15	4.47	35.96	4.05
	MCAv (cm/s)	58.38	6.51	56.53	9.58
Task	HR * (bpm)	87.52	11.91	87.57	13.70
	ABP * (mmHg)	88.47	11.57	95.33	17.84
	P_ET_CO_2_ * (mmHg)	48.12	3.13	46.08	3.22
	MCAv * (cm/s)	82.71	13.50	75.59	11.53

Note: HR: heart rate; ABP: arterial blood pressure; P_ET_CO_2_: partial pressure of end-tidal CO_2_; and MCAv: middle cerebral artery flow velocity. (*) statistically significant differences between basal and task conditions in the entire sample. No significant difference in any of the variables was observed between the two groups.

**Table 2 brainsci-12-00558-t002:** Derived Doppler variables.

Task	Variable	Med-Lows		Med-Highs	
		Mean	SD	Mean	SD
*HVT*	∆CVCi (cm/s/mmHg)	−0.32	0.12	−0.33	0.07
	CVR (cm/s/mmHg)	1.81	0.46	1.91	0.49
*RBT*	∆CVCi (cm/s/mmHg)	0.18	0.12	0.16	0.13
	CVR (cm/s/mmHg)	2.49	0.76	2.17	1.66

Note: CVCi: cerebrovascular conductance index; CVR: cerebrovascular reactivity.

## Data Availability

Data available upon request.

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
