# Peer review of "Cerebral Blood Flow in Healthy Subjects with Different Hypnotizability Scores"

_brainsci, 2022, doi:10.3390/brainsci12050558_

Round 1

Reviewer 1 Report

The authors of the manuscript “Hypnotizability-related Cerebral Blood Flow” investigate an important and interesting topic. A few issues remain to be addressed:

  1. The derivation of hypotheses is not clear, especially the correlations that were statistically significant according to the results.
  2. Sample size calculation is missing. What is the justification to test 24 participants? Why were participants not selected according to their suggestibility so that there are the same number of high and low suggestible participants in the study? As the comparison between high and low suggestible participants is the main topic of the study, I would expect that suggestibility was assessed first and then, the same number of high and low participants are included in the study. The authors should explain their procedure in more detail here.
  3. Table 1 is rather confusing. The asterisks for significance are indicating a statistical significance of baseline vs. task for the whole sample, while the columns suggest there are statistically significant differences between high and low suggestible participants. Please make that clearer.
  4. In the discussion section, it is not clear what the results mean in a broader context. The notion that high suggestible participants have “different metabolic requests” should be elaborated in more detail.

Author Response

To simplify the draft, as requested by Reviewers 1 and 3, we have erased a few parts (near-infrared spectroscopy – NIRS variables, comparisons between basal doppler variables) and a figure (original Fig. 5) and have clarified the significance of the two approaches (ANOVA and correlational analyses) by changing the subtitles in the Results and Discussion sections and by summarizing the main findings of each Results paragraph in their initial lines.

REVIEWER 1

The authors of the manuscript “Hypnotizability-related Cerebral Blood Flow” investigate an important and interesting topic. A few issues remain to be addressed:

The derivation of hypotheses is not clear, especially the correlations that were statistically significant according to the results.

The decision to correlate ABP and PETCO2 with MCAv was due to physiological considerations, as MCAv depends (in different amounts and in different conditions) on ABP and PETCO2. The correlation between MCAv and the derived variables CVCi and CVR was based on similar physiological considerations.

Sample size calculation is missing. What is the justification to test 24 participants?

As we mentioned in the Methods section of the original draft of the manuscript, the studied subjects had been enrolled in a previous study. After that, the instrumentation was no longer available, thus we could not enroll further participants.

Sample Size Calculation is based on: Confidence Interval (CI) of 95% with a Margin of Error (E) of 5% and Middle Cerebral Artery Flow Velocity (MCAv) Standard Deviation (σ) of 12.5.

n

n

n

n = (4.9) 2

n = 24.01

Therefore, the sample size calculation indicated a minimum number of 24 subjects and we decided to utilize the previously recorded measurements for the present study. Unfortunately, 2 participants could not be included in the analysis owing to bad signals. The derived variables CVR and CVCi would require a larger sample, thus we did not compare them between groups.

Why were participants not selected according to their suggestibility so that there are the same number of high and low suggestible participants in the study?

Regarding subjects’ selection, please see above (pre-recorded participants).

As the comparison between high and low suggestible participants is the main topic of the study, I would expect that suggestibility was assessed first and then, the same number of high and low participants are included in the study. The authors should explain their procedure in more detail here.

Not highs and lows, but med-highs and med-lows were in the same number. The decision to compare med-highs and med-lows rather than highs, mediums, and lows, as reported in other studies was made to increase the statistical power of the comparisons.

Table 1 is rather confusing. The asterisks for significance are indicating a statistical significance of baseline vs. task for the whole sample, while the columns suggest there are statistically significant differences between high and low suggestible participants. Please make that clearer.

Thank you for pointing this out: An asterisk (*) was mistakenly inserted in basal TOI in the Rebreathing (RBT) and we completely removed the NIRS parameters to simply the draft for better clarity. We also made the meanings of comparison clearer in the legend, by adding: No significant difference in any of the variables was observed between the two groups.

In the Discussion section, it is not clear what the results mean in a broader context. The notion that high suggestible participants have “different metabolic requests” should be elaborated in more detail.

Possibly different metabolic demands are dealt with in the Introduction (section 1.2. Changes in the cerebral arteries diameter induced by brain activity). They are referred to as hypnotizability-related differences in information processing suggested by topological studies of EEG during sensory and cognitive tasks [Reference 48]. This concept is again cited in the Discussion (section 4.1. Comparisons between groups) in which we hypothesize that hypnotizability-related information processing may account for the absence of MCAv differences in both groups during trail making task (TMT) and mental arithmetic task (MAT).

At the end of section 4.2. Association between systemic and Doppler variables; we have added a few lines to elaborate: During both hyperventilation and rebreathing, an association of cerebrovascular reactivity (CVR, related to PETCO2) and cerebrovascular conductance index (CVCi, related to ABP) was found only in med-lows. We hypothesize that a different interaction between arterial blood pressure and CO2 [Reference 37] occurs in med-highs and med-lows owing to the presence of different local metabolic conditions depending on different modes of information processing [Reference 9].

Reviewer 2 Report

In this paper, the authors investigate a fundamental issue in the hypnosis literature: hypnotizability. This study investigates how the MCAv is affected by the degree of hypnotizability while participants perform several tasks (cognitive tasks, hyperventilation and rebreathing). In addition, the authors wish to investigate how the MCAv is associated with partial pressure of end-tidal CO2 of tissue oxygenation index and arterial blood pressure in participants with different hypnotizability. The manuscript is very well written, the conclusions of the authors are in line with the experimentation that was conducted. Although the authors do not find the same results as other authors, the discussion and limitation section provides solid leads to explain this difference.

Introduction
The introduction is very well written, pleasant to read, well referenced.

Materials and methods

How did the authors check for sleep and attention disturbance? 

I note that the authors took the necessary precautions to reduce the variability linked to the experimental context (time of day, temperature, food). This is unfortunately rarely done in studies.

Can you give more details on how you did this? "Since hypnotizability is a stable individual trait [3] recordings performed 2 months earlier than hypnotic assessment could be reliably studied as a function of hypnotizability."

Can you give more details about the training phase of the participants? This could help the reproducibility of this study. "Before starting the tests sequence, participant received instructions and training to ensure that they were able to perform the tasks and respiratory maneuvers correctly."

Can you confirm that the order of the tasks performed by the participants was randomised?

Author Response

In this paper, the authors investigate a fundamental issue in the hypnosis literature: hypnotizability. This study investigates how the MCAv is affected by the degree of hypnotizability while participants perform several tasks (cognitive tasks, hyperventilation and rebreathing). In addition, the authors wish to investigate how the MCAv is associated with partial pressure of end-tidal CO2 of tissue oxygenation index and arterial blood pressure in participants with different hypnotizability. The manuscript is very well written, the conclusions of the authors are in line with the experimentation that was conducted. Although the authors do not find the same results as other authors, the discussion and limitation section provide solid leads to explain this difference.

Introduction: The introduction is very well written, pleasant to read, and well referenced.

Thank you for your kind appreciation.

Materials and methods:

How did the authors check for sleep and attention disturbance?

We enquired with the subjects at the time of recruitment. The following sentence was added: as self-reported while signing the informed consent.

I note that the authors took the necessary precautions to reduce the variability linked to the experimental context (time of day, temperature, food). This is unfortunately rarely done in studies.

Thank you for your comment.

Can you give more details on how you did this? "Since hypnotizability is a stable individual trait [3] recordings performed 2 months earlier than hypnotic assessment could be reliably studied as a function of hypnotizability."

This has been modified as follows for better clarity: Since hypnotizability is a dispositional trait substantially stable through life [Reference 3].

Can you give more details about the training phase of the participants? This could help the reproducibility of this study. "Before starting the tests sequence, the participant received instructions and training to ensure that they were able to perform the tasks and respiratory maneuvers correctly."

We modified the draft as follows: Participants were briefly familiarized with the tasks and respiratory maneuvers, particularly with the visual feedback during hyperventilation.

Can you confirm that the order of the tasks performed by the participants was randomized?

We confirm that: the test sequence was randomized for the different subjects. The sentence is slightly modified to better address this issue.

Reviewer 3 Report

Title
The title is incomprehensible and should be modified

Abstract
The conclusions should be presented more clearly

Materials and Methods
It is not clear how many subjects participated, 24 are mentioned but only 12 females are indicated: "Twenty-four healthy subjects (12 females; age: 26.1±4.5 years)."

This is a very interesting study. However, so many facts and topics have been dealt with that the manuscript seems overloaded. Both the introduction, the results and discussion section are too long and contain so much information that it is difficult to read the paper. Since the topic and the paper are well worth reading, it is recommended to focus on fewer parameters and shorten the manuscript significantly. The conclusions should be stated more clearly.

Author Response

To simplify the draft, as requested by Reviewers 1 and 3, we have erased a few parts (near-infrared spectroscopy – NIRS variables, comparisons between basal doppler variables) and a figure (original Fig. 5) and have clarified the significance of the two approaches (ANOVA and correlational analyses) by changing the subtitles in the Results and Discussion sections and by summarizing the main findings of each Results paragraph in their initial lines.

Title: The title is incomprehensible and should be modified.

The title has been modified as follows: Cerebral Blood Flow in Healthy Subjects with Different Hypnotizability Scores

Abstract: The conclusions should be presented more clearly.

The Conclusion has been modified as follows: For the first time, cerebrovascular reactivity related to hypnotizability was investigated, evidencing different correlations among hemodynamic variables in med-high and med-low subjects.

Materials and Methods

It is not clear how many subjects participated, 24 are mentioned but only 12 females are indicated: "Twenty-four healthy subjects (12 females; age: 26.1±4.5 years)."

Thank you for pointing this out. The sentence has been modified as follows: Twenty-four healthy subjects (age: 26.1 ± 4.5 years; 12 females and 12 males).

This is a very interesting study. However, so many facts and topics have been dealt with that the manuscript seems overloaded. Both the introduction, the results and the discussion section are too long and contain so much information that it is difficult to read the paper. Since the topic and the paper are well worth reading, it is recommended to focus on fewer parameters and shorten the manuscript significantly.

We have erased the information regarding NIRS parameters from the manuscript (Introduction, Methods, Results, and Discussion), as in fact, the analysis of tissue oxygenation index (TOI) did not provide relevant information to achieve the specific study aims.

The conclusions should be stated more clearly.

The Conclusion has been modified as follows: . Finally, it is noticeable that at variance with highs and lows, med-highs and med-lows represent half of the general population each. Thus, in a general perspective, the cerebrovascular reactivity of half of the population is expected to be more sensitive to the changes in the systemic blood pressure and the other half to the local increases in CO2 following the changes in blood pressure. Whether the different vascular reactivity to ABP changes of med-highs and med-lows is related to a different involvement of NO in the regulation of the vascular response remains to be ascertained. It may be speculated that this difference could influence the capacity to adapt to cerebrovascular insults.

We also revised the Graphical Abstract as well as the Figure 1. Looking forward to hearing from you soon. Thank you for the consideration of this manuscript.

Round 2

Reviewer 3 Report

All points have been sufficiently addressed. The paper can be recommended for publication.